# Development of the Sterile Insect Technique to control the dengue vector *Aedes aegypti* (Linnaeus) in Sri Lanka

**Tharaka Ranathunge[1,2], Jeevanie Harishchandra[3], Hamidou Maiga[4], Jeremy Bouyer[4] Y. I. Nilmini Silva Gunawardena[1], Menaka Hapugoda**[1] *

**1** Molecular Medicine Unit, Faculty of Medicine University of Kelaniya, Colombo, Sri Lanka, **2** Department of Biomedical Sciences, Faculty of Health Sciences, CINEC Campus, Malabe, Sri Lanka, **3** Anti-Malaria Campaign (AMC) Public Health Complex, Ministry of Health, Colombo, Sri Lanka, **4** Insect Pest Control Subprogramme, Joint FAO/IAEA Centre of Nuclear Techniques in Food and Agriculture, Department of Nuclear Sciences and Applications, International Atomic Energy Agency, Vienna, Austria

* menakaha@kln.ac.lk

**Data Availability Statement:** All relevant data are within the paper and its Supporting Information files.

## Abstract

### Background

The Sterile Insect Technique (SIT) is presently being tested to control dengue in several countries. SIT aims to cause the decline of the target insect population through the release of a sufficient number of sterilized male insects. This induces sterility in the female population, as females that mate with sterilized males produce no offspring. Male insects are sterilized through the use of ionizing irradiation. This study aimed to evaluate variable parameters that may affect irradiation in mosquito pupae.

### Methods

An *Ae. aegypti* colony was maintained under standard laboratory conditions. Male and female *Ae. aegypti* pupae were separated using a Fay and Morlan glass sorter and exposed to different doses of gamma radiation (40, 50, 60, 70 and 80 Gy) using a Co[60] source. The effects of radiation on survival, flight ability and the reproductive capacity of *Ae. aegypti* were evaluated under laboratory conditions. In addition, mating competitiveness was evaluated for irradiated male *Ae. aegypti* mosquitoes to be used for future SIT programmes in Sri Lanka.

### Results

Survival of irradiated pupae was reduced by irradiation in a dose-dependent manner but it was invariably greater than 90% in control, 40, 50, 60, 70 Gy in both male and female *Ae. aegypti*. Irradiation didn't show any significant adverse effects on flight ability of male and female mosquitoes, which consistently exceeded 90%. A similar number of eggs per female was observed between the non-irradiated groups and the irradiated groups for both irradiated males and females. Egg hatch rates were significantly lower when an irradiation dose above 50 Gy was used as compared to 40 Gy in both males and females. Irradiation at

**Funding:** Technical co-operation by the International Atomic Energy Agency (TC-SRL5/047, INT5155, RAS-50/82) and funding by the National Research Council, Sri Lanka (TO 14-04). The funders had no role in study design, data collection and analysis, decision to publish, or preparation of the manuscript.

**Competing interests:** The authors have declared that no competing interests exist.

**Abbreviations:** DW, Distilled Water; ANOVA, Analysis of Variance; SIT, Sterile Insect Technique; SPSS, Statistical Package for the Social Sciences; WHO, World Health Organization.

higher doses significantly reduced male and female survival when compared to the non-irradiated *Ae. aegypti* mosquitoes. Competitiveness index (C) scores of sterile and non-sterile males compared with non-irradiated male mosquitoes under laboratory and semi-field conditions were 0.56 and 0.51 respectively at 50 Gy.

## Signification

Based on the results obtained from the current study, a 50 Gy dose was selected as the optimal radiation dose for the production of sterile *Ae. aegypti* males for future SIT-based dengue control programmes aiming at the suppression of *Ae. aegypti* populations in Sri Lanka.

## Introduction

Mosquitoes transmit several pathogens, which are the causative agents of major diseases to humans such as dengue fever, chikungunya, Zika, Japanese Encephalitis, malaria, filariasis, etc [1, 2]. Among these diseases, dengue is presently the most important mosquito-borne viral infection in South-East Asia, including Sri Lanka [3]. With a change of serotype(s), a dramatic increase in the incidence of dengue and its severe manifestations created an alarming situation in Sri Lanka in 2017, with 185,686 cases recorded—the highest ever in a single year—and over 440 deaths [4].

Due to the absence of effective drugs or a vaccine for dengue serotypes, patient management and vector control are the most effective options to control dengue in the country [5]. In Sri Lanka, *Ae. aegypti* (Linnaeus) remains the primary vector, followed by *Ae. albopictus* (Skuse) as the secondary vector [6, 7]. Insecticide-based conventional tools are extensively used to control adult and larval mosquito populations. Such tools include chlorinated hydrocarbons like Dichloro-Diphenyl-Trichloroethane (DDT) or Dieldrin, organophosphates like diazinon or bensolid, and carbamates like carbofuran and carboxyl [8]. However, the use of insecticides is decreasing due to the development of insecticide resistant strains and unintended side effects on human and ecosystem health [9–12]. Therefore, the use of chemical insecticides is no longer considered to be an effective solution to suppress the vector population. New vector control approaches such as the Sterile Insect Technique (SIT), the Incompatible Insect Technique (IIT) and the use of transgenic mosquitoes have been developed recently as alternatives [13, 14].

SIT is a species-specific, non-polluting and eco-friendly method that has been used since 1977 to control insect pests [15]. This technique involves the release of large numbers of irradiated sterile male insects into the environment to compete with wild fertile males and mate with wild female insects. Male insects are exposed to gamma-irradiation, causing large scale random damage to the insect chromosomes and/or dominant lethal mutations in the sperm [16, 17]. Sterile males cannot become established in the environment, therefore the continuous release of artificially-reared sterile males is necessary to suppress or eliminate the population of vector or pest insects [18].

Raymond Bushland and Edward Knipling developed the SIT to eliminate screwworms preying on warm-blooded animals, especially cattle. The screwworm fly (*Cochliomyia hominivorax*) has successfully been eradicated from North and Central America through the use of this technique [19]. SIT is used in a number of countries to control fruit fly pests, particularly the Mediterranean fruit fly (*Ceratitis capitata*) and the Mexican fruit fly (*Anastrepha ludens*) [20]. Recent progress on the development of the SIT package against mosquitoes allows envisaging its deployment in several countries [21, 22]. For example, the National Institute for Public Health

(INSP) in Mexico is evaluating the possible use of SIT as an additional control measure for local strains of *Ae. aegypti* and *Ae. albopictus* in the most impacted areas of the country [23].

Developing SIT for the *Ae. aegypti* population is important for the integrated control of dengue in Sri Lanka. Optimizing the minimum dose to irradiate male mosquitoes is a prerequisite in the SIT approach. No attempt has yet been made in Sri Lanka to evaluate if and how survival, flight ability, reproductive capacity and competitiveness changes following male insect irradiation. Therefore, a proper evaluation should be conducted on such parameters of *Ae. aegypti* vector under laboratory and semi-field conditions, prior to the application of this technique in the country [21]. The present study was focused on evaluating the effects of different doses of irradiation on survival, flight ability, reproductive capacity and sexual competitiveness of sterile males as compared to wild males from local strains of *Ae. aegypti* under laboratory and semi-field conditions with the intention of future use of SIT to control dengue vectors in Sri Lanka.

## Materials and methods

### Establishment of an *Ae. aegypti* colony

Adult mosquito surveillance was conducted in the Ragama Public Health Inspector (PHI) division, located within the District of Gampaha, Sri Lanka in 2018, and captured mosquitoes were transported to the laboratory for colonization and mass-rearing at the Molecular Medicine Unit, Faculty of Medicine, University of Kelaniya, Sri Lanka. Within the laboratory, *Ae. aegypti* mosquitoes were isolated through morphological identification by well-trained entomologists. Eggs laid by single *Ae. aegypti* blood-engorged females were used to establish a mosquito colony of *Ae. aegypti*. Eggs were mixed thereafter to build a colony that colony was maintained in 24 × 24 × 24 cm cages with mesh screening on top, under a 12:12 (light:dark) cycle at standard conditions (at 27 ± 2°C and 75 ± 5% humidity).

Eggs collected from mosquitoes established at the insectary in the Molecular Medicine Unit, Faculty of Medicine, University of Kelaniya, Sri Lanka were transferred to 1 L plastic trays. Hatched larvae were counted and transferred to a properly labeled plastic tray (40 cm × 30 cm × 5 cm) containing 2 L of distilled water. For the purposes of quality control and to maintain uniform size of the larvae, a density of 1,000 larvae per tray was used. Initially, larvae were fed with 1.5 mL of the International Atomic Energy Agency (IAEA) recommended larval diet [24]. The larval diet was added once a day to a tray according to the following regime: day 1, 1.5 mL; day 2, 1.55 mL; day 3, 1.6 mL; day 4, 1.65 mL; and day 5, 1.7 mL. Once larvae became pupae, they were transferred to 500 mL plastic cups containing distilled water [24, 25].

After the emergence of adult mosquitoes, adults were counted and transferred into adult rearing cages (30 cm × 30 cm × 30 cm). About 700 adult mosquitoes were reared in one cage with a 1:1 male to female ratio. Adult mosquitoes were fed with 10% sucrose solution. Females were given a blood meal of cattle origin using the Hemotek (PS-6 System, Discovery Workshops, Accrington, UK) artificial membrane feeder [26]. The feeder was then placed on top of adult cages which allowed female mosquitoes to feed for around 1–2 hrs. After blood feeding, sugar cups were placed inside the adult cages. Then, 48 hrs after blood feeding, egg collection was done by placing an egg collection cup (250 mL) containing 10 mL of distilled water and cotton and egg laying filter paper inside the cages for about two days. Egg papers were then removed from the cages and kept in the cups for an additional 24 hrs [26].

### Sex separation

Male and female *Ae. aegypti* pupae were separated using a Fay-Morlan glass plate sorter (M5412, John W. Hock Company, Gainesville, USA) [27].

## Dose mapping

Male and female pupae aged 24 hrs were irradiated separately using an irradiator (Gammacell 220, Atomic Energy of Canada Ltd., Co[60]) located in The Horticultural Crop Research and Development Institute, Sri Lanka in dry conditions. Dosages were determined using the Fricke dosimetry system [28]. Five different radiation doses (40, 50, 60, 70 and 80 Gy) were used for dose mapping, and controls without exposure to irradiation were maintained for all experiments.

## Effects of irradiation on pupal mortality

After irradiation, 150 pupae were transferred to 250 mL plastic cups containing distilled water, and the pupae in these cups were placed inside acrylic cages (30 cm × 30 cm × 30 cm) and left to emerge. After 72 hrs, dead pupae in plastic cups were counted and the total survival of adults was calculated. Three replicates were performed for each dose and each sex.

## Effects of irradiation on flight ability

To determine the flight ability of adults, 150 pupae of one sex were placed in a 9 cm diameter petri dish into which a transparent tube (25 cm in height and 8 cm in diameter), was introduced. This tube was placed inside an adult cage (30 cm × 30 cm × 30 cm). Flight ability was measured considering the percentage of emerged adults from the pupae that could exit the tube over a 48h period. This procedure was performed three times for both sexes at all doses.

## Effects of irradiation on fecundity and fertility

After the sterilization process, 100 irradiated male *Ae. aegypti* aged 4–5 days were mated with fertile females at a 1:1 ratio in adult mosquito cages (30 cm × 30 cm × 30 cm) under standard laboratory conditions. Adult mosquitoes were fed with cattle blood using Hemotek membrane feeders. Two days (48 hrs) after blood feeding, egg collection cups (250 mL) internally coated with filter paper and half-filled with water were placed into the cages. Female mosquitoes were allowed to lay eggs for about 48 hrs. Egg papers were then removed from the cages and kept in the cup for an additional 24 hrs. The number of eggs laid in each ovitrap (fecundity) was counted using a binocular dissecting microscope (Olympus Optical Co.Ltd., Tokyo). Collected eggs were allowed to hatch (fertility) and larvae were counted. To determine the effects of irradiation on females, the experiment was repeated using 100 irradiated females and 100 virgin males that had not been irradiated. Controls consisted of groups of 100 males and 100 females that had not been irradiated. Three replicates (cages) were performed for each experiment.

## Effects of irradiation on adult survival

For each dose, groups of 25 males and 25 females were placed separately in adult mosquito cages (30 cm × 30 cm × 30 cm). The mosquitoes were provided with continuous access to 10% sucrose solution. Mortality was recorded daily until the death of the last individual. Three replicates (cages) were performed for both sexes and each dose.

## Competitiveness index (C) of sterile male mosquitoes

Based on the results obtained from dose mapping experiments, an optimal irradiation dose inducing nearly 100% sterility (50 Gy) was selected to estimate the competitiveness index (C) [28] under laboratory and semi-field conditions.

**Laboratory conditions.** Four- to five-day-old sterilized and untreated males were introduced into adult mosquito cages (30 cm × 30 cm × 30 cm) together with virgin females of the

same age at 1:1:1, 3:1:1 and 5:1:1 ratios (sterile males to untreated males to virgin females) (n = 100). Furthermore, a 1:1 ratio of sterile males to virgin females and a 1:1 ratio of untreated males to virgin females were introduced into separate cages. Adult mosquitoes were fed with cattle blood and fed females were counted. Blood-fed females were allowed to lay eggs and the number of eggs laid in each egg cup were counted. Collected eggs were hatched and larvae were counted.

**Semi-field conditions.** Three experimental ratios of 1:1:1, 3:1:1 and 5:1:1 (sterile males to fertile males to virgin females, n = 100) were tested in semi-field cages (1.82 m × 1.21 m × 1.21 m) four to five days after emergence. Controls were set at a 1:1 ratio of sterile males to virgin females and a 1:1 ratio of untreated males to virgin females. After a mating period of three days, females were collected into cages (30 cm × 30 cm × 30 cm) and brought to the laboratory. Mosquitoes were blood fed and egg hatch rates were compared as done previously.

## Statistical analysis

All statistical analyses were performed in R (version 4.0.2; https://cran.r-project.org) using RStudio (RStudio, Inc.Boston, MA, USA, 2016). Binomial linear mixed effect models were used to analyze the impact of irradiation on pupal survival and flight ability. Repetitions were treated as random effects and the treatment levels for irradiation were used as fixed effects. Because they were not normally distributed, egg hatch rates were arcsin-sqrt-transformed prior to analysis as a function of irradiation doses with linear mixed model fit by REML with egg hatch rates as response variable and dose, sex and their interaction considered as fixed effects. Multiple comparisons of mean egg hatch rates between doses were performed using the emmeans function of the emmeans package [29]. The mosquito survivorship was analyzed as a function of the irradiation treatment and dose using the Cox mixed effects model fit by maximum likelihood ("coxme" function in the "survival" package) [30]. The number of eggs per female was analyzed with Poisson errors. Irradiation doses (5 levels: 40, 50, 60, 70, 80 Gy), control (untreated) and sex (2 levels: male, female) were considered as fixed factors and replicate (cage) as random factor.

For the validation, the full models were checked for overdispersion [31] and for normality and homogeneity of variances on the residuals [32]. Model simplification used the stepwise removal of terms, followed by likelihood ratio tests (LRTs). Term removals that significantly reduced explanatory power (p < 0.05) were retained in the minimal adequate model [33]. Differences between the levels of significant fixed factors were analyzed using post hoc Tukey tests (glht function in package multcomp) [34]. The significant interactions were analyzed using the emmeans function (in package emmeans) [29].

Fried competitive index 'C', defined by Fried (1971) [2] was calculated for laboratory cages and semi field cages using hatch rates of the fertile and sterile control groups, and the observed egg hatch rates with a 1:1:1 ratio of sterile males to wild males to wild females in experimental cages. The formula (Hn-Ho) / (Ho-Hs))*(N/S) was used to calculate the Fried Competitiveness index (C), where Hn and Hs denote hatch rate from eggs of females mated with untreated and sterile males, respectively, Ho is the observed egg hatch rate under a 1:1:1 ratio, N is the number of untreated males and S is the number of sterile males.

## Results

### Effects of irradiation on pupal survival

The highest and lowest pupal survival up to the adult stage for male *Ae. aegypti* were 96.22% in control and 81.11% at 80 Gy, respectively. There was no significant difference among the percentage of pupal survival up to the adult stage for both male and female *Ae. aegypti* with

irradiation doses of 0 to 40 Gy (P > 0.05). Percentage pupal survival up to the adult stage significantly varied among different irradiation doses (p<0.05) (Fig 1). Although adult emergence rates were higher than 90%, fewer adults emerged from 50 to 80 Gy irradiated pupae as compared to the non-irradiated pupae. Emergence rates were significantly higher for females than males (p<0.05) (Fig 1).

## Effects of irradiation on flight ability

The percentage scores for flight ability for control to irradiation treatments of control and irradiation treatments were 98.84% to 96.99% for males and 99.09% to 94.05% for females respectively. Similar escape rates were observed between non-irradiated mosquitoes and those that were irradiated with 40 Gy (P > 0.05) for both male and female *Ae. aegypti*. Although adult escape rates were higher than 90%, lower adult escape rates were observed in all irradiated mosquitoes groups as compared to the non-irradiated mosquitoes for both males and females. Male mosquito escape rates were similar to that seen in female mosquitoes (P > 0.05) (Fig 2).

## Effects of irradiation on female fecundity and fertility

Regardless of the cross type between irradiated males and non-irradiated females (Fig 3) or between non-irradiated males and irradiated females (Fig 4), no impact of irradiation was observed on mosquito fecundity (P > 0.05). Similar numbers of eggs per female of were observed between the non-irradiated groups and the irradiated groups.

The fertility was reduced from 87.66% in control to zero in 80 Gy irradiated male *Ae. aegypti* mated with non-irradiated females (Fig 5A). Similarly, in irradiated females that mated with non-irradiated males, fertility was 92.53% in control groups and was reduced to zero in treatments involving 70–80 Gy (Fig 5B). Irradiation induced higher sterility as compared to the non-irradiated control (p<0.05). Egg hatch rates were lower (p < 0.05) with doses above 50Gy as compared to 40Gy. No difference in egg hatch rates was observed when male or female pupae were irradiated (P > 0.05).

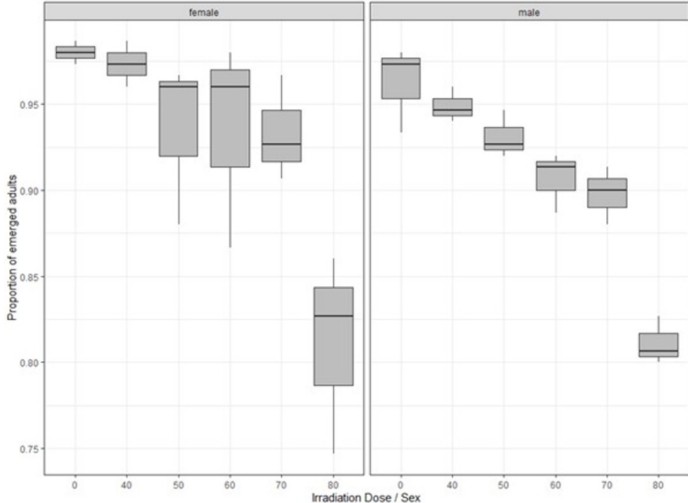

**Fig 1. Percentage of pupal survival up to adult stage of male and female *Ae. aegypti* following exposure to different irradiation doses.** Each box denotes the median as a line across the middle, the quartiles (25th and 75th percentiles), the minimum and maximum values at the ends of the vertical lines. Results are expressed as mean ± SE.

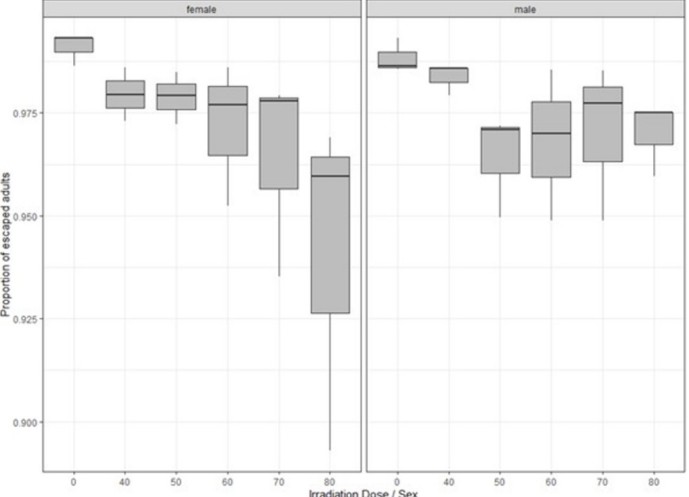

**Fig 2. Effects of irradiation on flight ability of male and female *Ae. aegypti* following exposure to different irradiation doses.** Each box denotes the median as a line across the middle, the quartiles (25th and 75th percentiles), the minimum and maximum values at the ends of the vertical lines. Results are expressed as mean ± SE.

## Effects of irradiation on adult survival

The survival curves of male and female *Ae. aegypti* irradiated with different doses are presented in Figs 5 and 6, respectively. The dose of 80 Gy reduced significantly male survival as compared to the control ($\chi^2$ = 19.07, df = 5, P < 0.05). The survival rates were not significantly different between non-irradiated and irradiated male mosquitoes with 40 to 70 Gy (P > 0.05)

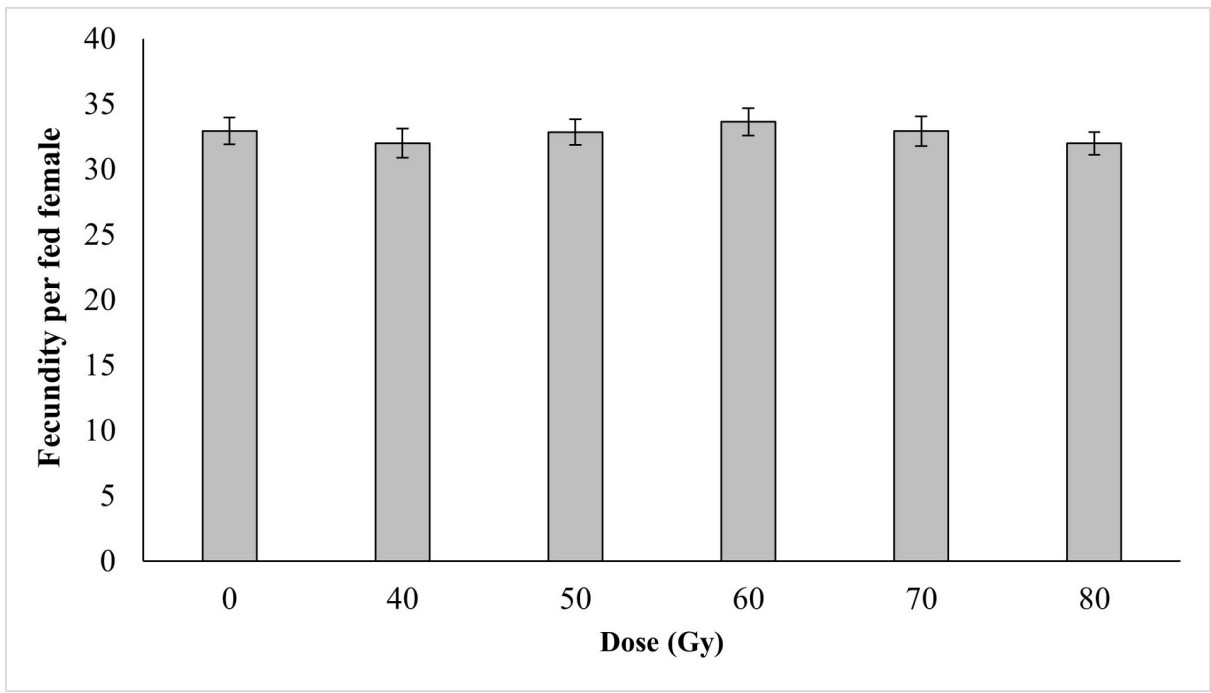

**Fig 3. Fecundity per fed female of fertile females matted with irradiated males *Ae. aegypti* following exposure to different irradiation doses.**

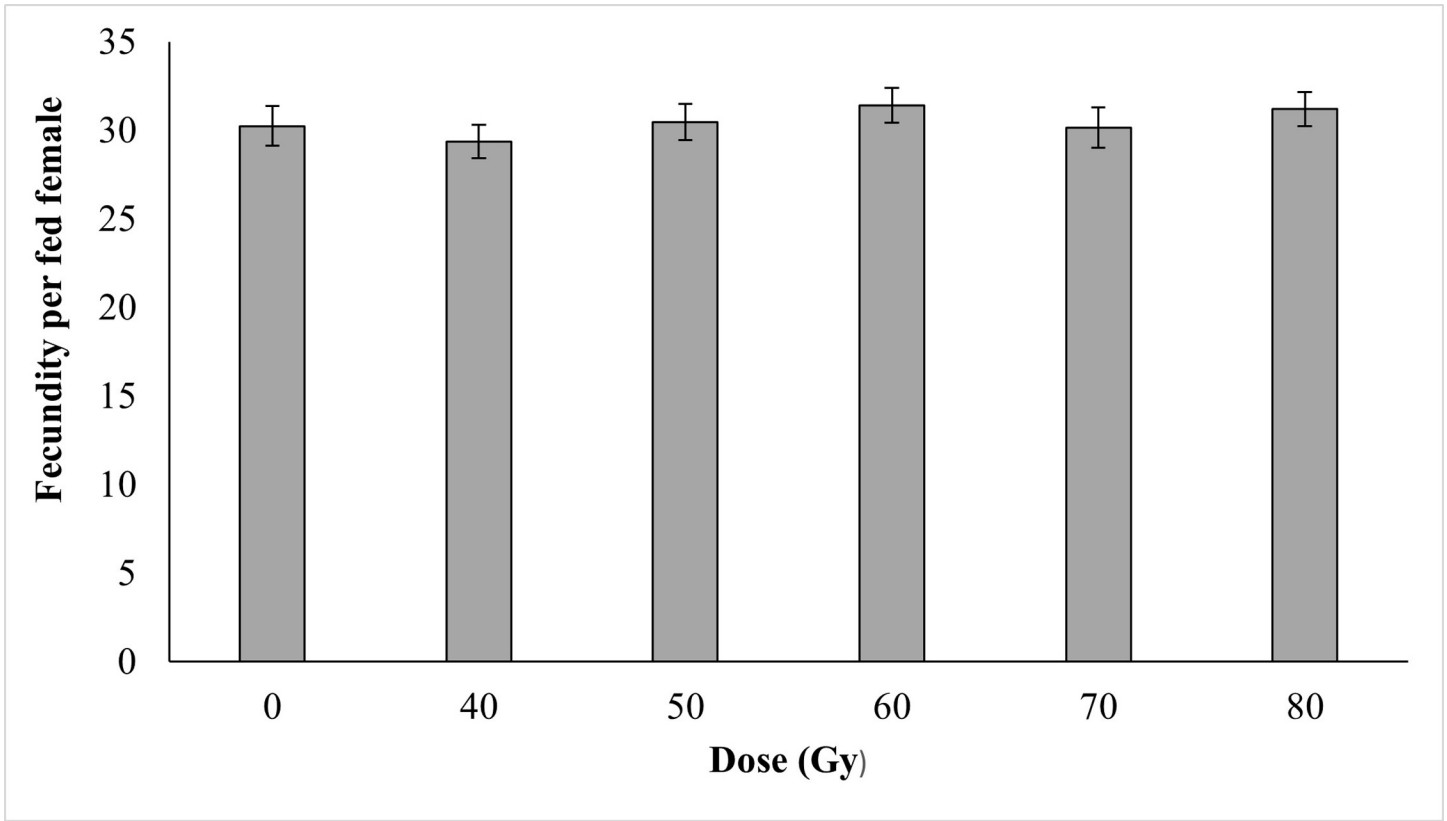

**Fig 4. Fecundity per fed female of fertile males matted with irradiated females *Ae. aegypti* following exposure to different irradiation doses.**

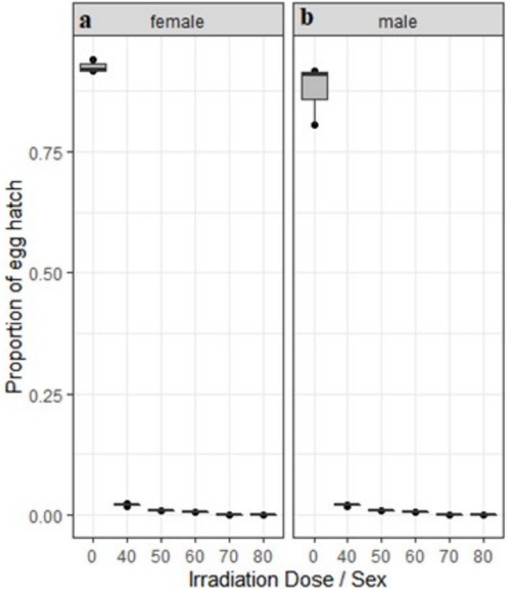

**Fig 5. Effects of irradiation on percentage of egg fertility of *Ae. aegypti*.** a Irradiated females mated with non-irradiated males. b Irradiated males mated with non-irradiated females. Each box denotes the median as a line across the middle, the quartiles (25th and 75th percentiles), the minimum and maximum values at the ends of the vertical lines. Results are expressed as mean ± SE.

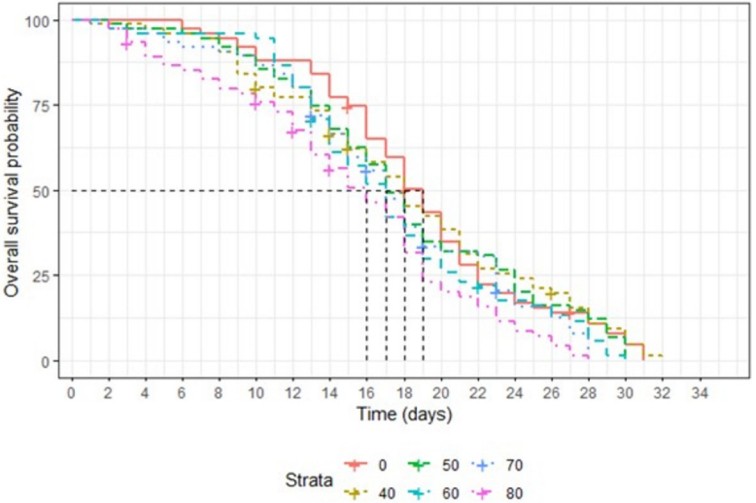

**Fig 6. Longevity of non-irradiated and irradiated with different doses of male *Ae. aegypti*.**

(Fig 6). Doses of 60, 70 and 80 Gy induced significant effects on female survival as compared to the non-irradiated groups ($\chi^2$ = 13.03, df = 5, P < 0.05) (Fig 7). There was no significant difference of adult longevity among non-irradiated and irradiated females at doses of 40 and 50 Gy (P > 0.05).

## Competitiveness index (C) of sterile male mosquitoes

The Hatch rates (%) were significantly different between irradiated and untreated controls (Tukey's posthoc test: all P <0.05). However, hatch rates (%) of 1:1:1 ratio were not significantly different from the 3:1:1 ratio (P > 0.05) but it was significantly different from the 5:1:1 (p<0.05). The competitiveness index (C) of sterilized males was 0.56 meaning that sterile males were about half as competitive as untreated males under laboratory conditions (Table 1).

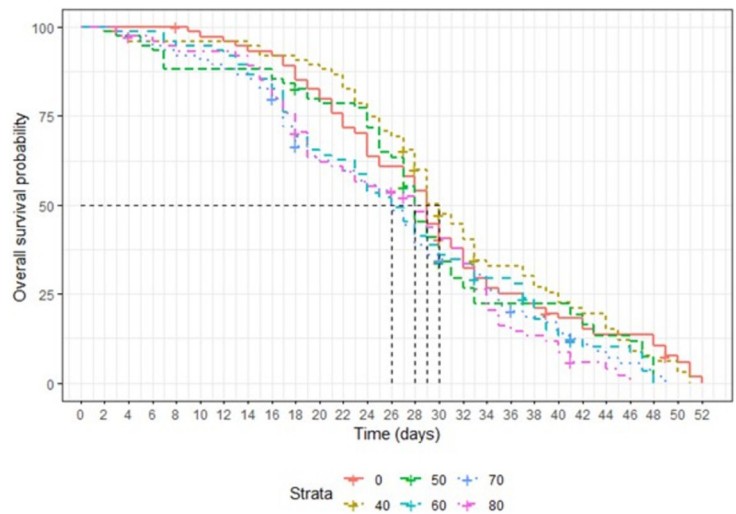

**Fig 7. Longevity of non-irradiated and irradiated with different doses of female *Ae. aegypti*.**

**Table 1. Competitiveness index (C) of sterile *Ae. aegypti* males under laboratory conditions measured with different ratios of sterile to untreated males.** Hatch rates (%) of females as a function of the different ratios (sterile males: untreated males: virgin untreated females). Different letters indicate significant differences between the ratios (Tukey's posthoc test P < 0.05). Hn and Hs are the hatch rate from eggs of females mated with untreated (untreated control) or sterile (sterile control) males respectively. Ho is the observed egg hatch rate for each ratio.

| Ratio | Hatch rates (%) | Competitiveness index (C) |
|---|---|---|
| Fertile control | 87.66(Hn)[a] | |
| Sterile control | 0.77(Hs)[b] | |
| 1:1:1 | 56.35(Ho)[c] | 0.56 |
| 3:1:1 | 47.66[c] | |
| 5:1:1 | 33.67[d] | |

Under semi-field conditions, hatch rates (%) were different between sterile and untreated controls (p<0.05). However, hatch rates were similar between 1:1:1 and 3:1:1 ratios (P > 0.05). The competitiveness index (C) of sterilized males was 0.51, meaning that sterile males were about half as competitive as untreated males under semi-field conditions (Table 2).

## Discussion

SIT-based vector control programmes involve area-wide integrated vector management including the release of a large number of sterile males to suppress the vector population. The current study was conducted to evaluate the impact of irradiation on the fertility of male and female *Ae. aegypti* under laboratory and semi-field conditions. The goal was to prepare the implementation of SIT at field level to suppress *Ae. aegypti* vector populations in order to reduce the risk of dengue transmission in Sri Lanka.

According to the results obtained from the current study, the survival of irradiated pupae was invariably greater than 90% in control, 40, 50, 60 and 70 Gy. In previous studies, radiation up to a 80 Gy dose did not significantly influence the survival of pupae of *Ae. aegypti* and *Ae. albopictus* up to the point of adult emergence [35]. *Anopheles arabiensis* mosquitoes were exposed at doses as high as 100 Gy without any adverse effect on pupal mortality [36].

In the current study, flight ability was evaluated using a tall tube developed for quality control of irradiated fruit flies [37]. Irradiation had no significant adverse effects on the flight ability (capacity to fly out of a test device) of *Ae aegypti* male and female mosquitoes, which consistently exceeded 90%. This might be due to an insufficient sensitivity of the flight device used in this study, since significant decreases in flight ability of adult male *Ae. aegypti* and *Ae. albopictus* mosquitoes was reported for lower doses than 70 Gy using a different device developed by the FAO-IAEA joint Subprogramme [38].

**Table 2. Competitiveness index (C) of sterile *Ae. aegypti* males under semi field conditions in large cages, measured with different ratios of sterile to untreated males.** Hatch rates (%) of females as a function of the different ratios (sterile males: untreated males: virgin untreated females). Different letters indicate significant differences between the ratios (Tukey's posthoc test P < 0.05). Hn and Hs are the hatch rate from eggs of females mated with untreated (untreated control) or sterile (sterile control) males respectively. Ho is the observed egg hatch rate for each ratio.

| Ratio | Hatch rates (%) | Competitiveness index (C) |
|---|---|---|
| Fertile control | 89.64(Hn)[a] | |
| Sterile control | 0.98(Hs)[b] | |
| 1:1:1 | 59.86(Ho)[c] | 0.51 |
| 3:1:1 | 51.35[c] | |
| 5:1:1 | 39.37[d] | |

Shetty et al. (2016) [39] reported that *Ae. aegypti* females that mated with irradiated males continued to lay eggs even when males had been exposed to doses in the range 10–300 Gy. Similarly, the egg production of irradiated males and females did not show any significant variation as compared to the control in the current study.

The fertility of females that mated with irradiated males was significantly reduced in *Ae. aegypti* at all doses, and zero fertility was observed at 80 Gy. The fertility of irradiated males was below 1% at 50 and 60 Gy. The fertility of irradiated *Ae. aegypti* females that mated with non-irradiated males was null at doses of 70 and 80 Gy, whereas the fertility was below 1% at 50 and 60 Gy, indicating no significant difference of fertility between irradiated males and females. Bond et al. (2019) [30] indicated that females were more susceptible to irradiation than males. Similarly, Shetty et al. (2016) [34] observed that fertility decreased after exposing male *Ae. aegypti* to 20–50 Gy. A negative correlation between egg fertility and irradiation dose has been reported across several species of mosquito [39–42].

Irradiating biological materials using ionizing radiation potentially damages the cells of insects [43], which can negatively affect adult performance [44–47]. According to the present experiment, high doses of radiation were detrimental to adult longevity in males. Similarly, when irradiated at doses beyond 80 Gy, the survival of *Anopheles* species was reduced [43, 44]. Other authors reported decreased insemination rates of sterile males and a reduced ability to find females or achieve successful copulation. The irradiation can increase the inactive periods of the fruit fly *Bactrocera tryoni* and reduce walking speed [45, 46]. Irradiation at high doses directly affects cell division (germ cells) through lethal mutations and also damages somatic cells, resulting in oxidative stress and cellular death [42]. Reduced longevity is often a result of radiation-induced somatic damage and this must be measured, ideally, under conditions that induce stress to emphasize any differences [42, 43]. Specifically, male survival during the first days of adult life is important as this is the period when mating is expected to occur after release [44].

Although the magnitude of the irradiation doses adversely impacts male mosquito quality and mating competitiveness, different irradiation doses were used to sterilize male mosquitoes for SIT applications in many countries [48, 49]. In Indonesia, 70 Gy was used for the sterilization process of male *Ae. aegypti* with of γ-ray sterilization, and there was no adverse effect on the mating ability and sterility of sterile male *Ae. aegypti* [49]. *Anopheles coluzzii*, one of the major vectors of malaria in sub-Saharan Africa, was sterilized at 90 Gy, and the egg hatch rate was significantly reduced to 20% [22]. However, excessive doses can reduce male quality [21]. Therefore, proper doses should be optimized by examining insect quality after irradiation as in the present study before implementation at field level.

Based on the results obtained from the dose optimization study under laboratory conditions, 50 Gy doses were used to evaluate the competitiveness of treated males mating with untreated females. Similarly, Bond *et al.* (2019) [35] showed that a sterilizing dose of 50 Gy resulted in little reduction in survival times of males, fertility was less than 1%, and no adverse effects on flight ability and fecundity of *Ae. aegypti* were reported. Therefore, they suggested that 50 Gy can be used for future evaluations of SIT-based control of *Ae. aegypti* in Mexico [36].

The Fried competitiveness index [50] was calculated for *Ae. aegypti* and showed that the sterile males were half as competitive as the wild untreated males under laboratory and semi-field conditions. That reduction in competitiveness is probably due to the variation of some parameters such as completive ability, sperm production and the flight ability of sterile males. In this study, a ratio of 5:1:1 was observed to have more impact on the hatch rates than the 3:1:1 and 1:1:1 ratios. The suppression efficacy of vector populations will depend on the level of induced sterility and release ratio [51, 52]. A study conducted in *An. arabiensis* showed that 81% sterility could be achieved by irradiating males at 75 Gy and increasing the sterile to untreated male ratio up to 10:1 [48].

The selection of developmental stage and age for irradiation of mosquitoes or other insects (having complete metamorphosis) is imperative due to the development of reproductive organs and maturity at different life stages [43]. In the process of mature sperm cell production, there are several steps, including primordial germ cells, primary and secondary spermatogonia, primary and secondary spermatocytes, spermatids and spermatozoa (mature sperm cells) [43]. Following this, mature sperm cells are released into the sperm reservoir located in the testis. The time taken for spermatogenesis is different from species to species and for mosquitoes it mainly occurs during the larval and pupal stages. When the male mosquitoes emerge from the pupal stage, spermatocytes continue maturing into spermatozoa and are released into the sperm reservoir. The testes are filled with 45% and 41% of mature sperm cells in newly emerged male of *An. stephensi* [52] and *An. culicifacies* [52], respectively. Therefore, the late pupal stage or the early adult stage should be considered most appropriate for irradiation. As an example, tephritid fruit flies are usually irradiated one or two days prior to adult emergence [53].

In conclusion, we suggest using 50 Gy to irradiate male *Ae. aegypti* to obtain 99% sterility with minimal adverse effects on adult survival. Before the release of sterile males into the environment, field level experiments such as mark-release-recapture are recommended to investigate the release rate of sterile males to wild untreated males into the environment in order to optimize the efficiency of SIT.

## Supporting information

**S1 Appendix.**
(PDF)

**S2 Appendix.**
(CSV)

**S3 Appendix.**
(CSV)

**S4 Appendix.**
(CSV)

**S5 Appendix.**
(PDF)

## Acknowledgments

Technical co-operation for the study was provided by the International Atomic Energy Agency (TC-SRL5/047, INT5155, RAS-50/82). Resources including man power and operating cost for the study were provided by the National Research Council, Sri Lanka (TO-14/04). Dr. Kostas Bourtzis Insect Pest Control Laboratory, Joint FAO/IAEA Division of Nuclear Techniques in Food and Agriculture, International Atomic Energy Agency for technical support for the development of the laboratories. Dr. W. L. G. Samarasinghe (Director) and Mr. R. M. J. C. B. Senanayake (Radiation Safety Officer), Horticulture Research and Development Institute, Department of Agriculture, Gannoruwa, Sri Lanka are acknowledge for their support.

## Author Contributions

**Conceptualization:** Menaka Hapugoda.

**Data curation:** Tharaka Ranathunge, Jeevanie Harishchandra.

**Formal analysis:** Tharaka Ranathunge.

**Funding acquisition:** Menaka Hapugoda.

**Methodology:** Tharaka Ranathunge.

**Software:** Tharaka Ranathunge.

**Supervision:** Y. I. Nilmini Silva Gunawardena, Menaka Hapugoda.

**Writing – original draft:** Tharaka Ranathunge.

**Writing – review & editing:** Hamidou Maiga, Jeremy Bouyer, Y. I. Nilmini Silva Gunawardena, Menaka Hapugoda.

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
