## [Decision Letter · Decision Letter 0]

2 Nov 2021

PONE-D-21-28806Development of Sterile Insect Technique (SIT) to control the dengue vector, Aedes aegypti (Linnaeus) in Sri LankaPLOS ONE

Dear Dr. Hapugoda,

Thank you for submitting your manuscript to PLOS ONE. After careful consideration, we feel that it has merit but does not fully meet PLOS ONE’s publication criteria as it currently stands. Therefore, we invite you to submit a revised version of the manuscript that addresses the points raised during the review process.

 Please submit your revised manuscript by Dec 17 2021 11:59PM. If you will need more time than this to complete your revisions, please reply to this message or contact the journal office at plosone@plos.org. Please include the following items when submitting your revised manuscript:A rebuttal letter that responds to each point raised by the academic editor and reviewer(s). You should upload this letter as a separate file labeled 'Response to Reviewers'.A marked-up copy of your manuscript that highlights changes made to the original version. You should upload this as a separate file labeled 'Revised Manuscript with Track Changes'.An unmarked version of your revised paper without tracked changes. You should upload this as a separate file labeled 'Manuscript'.

We look forward to receiving your revised manuscript.

Kind regards,

Ahmed Ibrahim Hasaballah

Academic Editor

PLOS ONE

Journal Requirements:

3. The statement 'The Ae. aegypti strain used in this experiment originated from eggs collected from the Ragama Public Health Inspector (PHI) division, located within the District of Gampaha, Sri Lanka in 2018' is not sufficient to determine the source of mosquitos in this study. In your Methods section, please provide additional details regarding the source of the mosquito eggs used in your study (if Ragama Public Health Inspector (PHI) division provided the eggs for free or as a donation please provide more information detailing this e.g. if the eggs are part of a insectary colony at the Ragama Public Health Inspector (PHI) division that were donated to this project, or e.g. the eggs are donated from Ragama Public Health Inspector (PHI) division in routine season surveillance of [provide name of area where surveillance occurs]) . If these was collected by members of this study please include geographic coordinates of your field collection site if available and provide the permits you obtained for the work. Please ensure you have included the full name of the authority that approved the field site access and, if no permits were required, a brief statement explaining why. If this was purchased then please include the names of the purchasing sources (e.g., stores, markets, suppliers), if available, as well as any further details about the purchased items (e.g., lot number, source origin, description of appearance) to ensure reproducibility of the analyses. For more information regarding PLOS' policy on materials sharing and reporting, see https://journals.plos.org/plosone/s/materials-and-software-sharing#loc-sharing-materials.

4. Please include your tables as part of your main manuscript and remove the individual files. Please note that supplementary tables (should remain/ be uploaded) as separate "supporting information" files

Reviewers' comments:

Reviewer's Responses to Questions

**Comments to the Author**

1. Is the manuscript technically sound, and do the data support the conclusions?

Reviewer #1: Yes

Reviewer #2: Yes

2. Has the statistical analysis been performed appropriately and rigorously? 

Reviewer #1: Yes

Reviewer #2: Yes

3. Have the authors made all data underlying the findings in their manuscript fully available?

Reviewer #1: Yes

Reviewer #2: Yes

4. Is the manuscript presented in an intelligible fashion and written in standard English?

Reviewer #1: Yes

Reviewer #2: Yes

5. Review Comments to the Author

Reviewer #1: This is a technical report about “Development of Sterile Insect Technique (SIT) to control the dengue vector, Aedes aegypti (Linnaeus) in Sri Lanka”.

The paper is clear and very well written, and gives insight into the effects of radiation on survival, flight ability and reproductive capacity of Ae. aegypti under laboratory conditions and the sterile male performance in field cage conditions for a better implementation of the future SIT programmes in Sri Lanka. The materials and methods are clearly indicated and the results reflect this. The results are also adequately discussed.

It is recommended that the manuscript be accepted for publication.

You may find some comments/suggestions below minor.

Authors affiliations:

4"Insect Pest Control Laboratory" instead "Insect Pest Control Subprogramme"

Abstract

Delete the space before the full-stop at the end of the 1st sentence ‘after Ae. Aegypti’.

Materials and methods

Effects of irradiation on flight ability

Remove the “.” after “9 cm diameter” in the sentence “…..150 pupae of one sex were placed in a 9 cm diameter. Petri dish into which a transparent tube…”

Statistical analysis

1. Why “the egg hatch rates were arcsinsqrt-transformed prior to analysis”?

This means that the data was not normally distributed? If the case please precise.

2. To assess the effects of irradiation on the different parameters (mortality, productivity), the author limits the number of replications to three, however is such studies more replicates are needed to have solid results.

Flies were not available in large number?

3. Mosquitoes -like many insects used for the mass-rearing for SIT- need to be checked for their quality control before using it for SIT. There are several quality control indicators that normally use for this purpose i.e. Mortality/survival, productivity, emergence rate, flight ability, mating competitiveness and insemination rate. In this study, to assess the sterile male mating performance in semi field cage, the author ignore some indicators as the propensity of mating, relative mating index and relative mating performance and limit the study to the competitiveness index.

Or with mosquitoes, is it difficult to evaluate these performance indicators?

4. What means “Hn” and “Hs” in the Friend formula “(Hn-Ho) / (Ho-Hs))*(N/S)”? Please define.

5. The author needs to submit the data analysis details in Rmarkdown file to make the follow up of the data is clear and simple.

Results

1. The figure 2 must be announced in the “Effects of irradiation on flight ability”

2. In the section “Effects of irradiation on adult survival”, please cheek the figure numbers

3. Section “Competitiveness index (C) of sterile male mosquitoes”:

- The tables 1 and 2 are announced but the tables are missing in the document.

- The author took about 5:1:1 ratio in the result section but is not mentioned in the methodology section

Reviewer #2: The manuscript shows a very encouraging beginning of SIT development for Sri Lanka. Great job of the IAEA supporting with technical expertise for this country scientific group. Overall, each of every section in the manuscript accomplishes with academic rules of experimental design; i.e, sample size, repetitions, lab rearing conditions, etc. Field research of mark-release-recapture should be started as soon as possible in order to settle preliminary releases of sterile males in Sri Lanka.

6. PLOS authors have the option to publish the peer review history of their article (what does this mean?). If published, this will include your full peer review and any attached files.

Reviewer #1: No

Reviewer #2: No

---

## [Author Response · Author response to Decision Letter 0]

17 Feb 2022

Reply to reviewers

Manuscript No: PONE-D-21-28806 

We would like to thank all the reviewers for their valuable comments, which contributed to improve the quality of the manuscript. We would like to emphasize that all suggestive changes have been addressed as much as possible. Changes made in the manuscript have been highlighted for your convenience. 

Journal Requirements

Comment 1

Reply

We have checked the guidelines and the manuscript is in the recommended format.

Comment 2

We suggest you thoroughly copyedit your manuscript for language usage, spelling, and grammar. If you do not know anyone who can help you do this, you may wish to consider employing a professional scientific editing service.

Reply

The manuscript has now been corrected for language usage, spelling, and grammar.

Comment 3

The statement 'The Ae. aegypti strain used in this experiment originated from eggs collected from the Ragama Public Health Inspector (PHI) division, located within the District of Gampaha, Sri Lanka in 2018' is not sufficient to determine the source of mosquitos in this study. In your Methods section, please provide additional details regarding the source of the mosquito eggs used in your study (if Ragama Public Health Inspector (PHI) division provided the eggs for free or as a donation please provide more information detailing this e.g. if the eggs are part of a insectary colony at the Ragama Public Health Inspector (PHI) division that were donated to this project, or e.g. the eggs are donated from Ragama Public Health Inspector (PHI) division in routine season surveillance of [provide name of area where surveillance occurs]) . If these was collected by members of this study please include geographic coordinates of your field collection site if available and provide the permits you obtained for the work. Please ensure you have included the full name of the authority that approved the field site access and, if no permits were required, a brief statement explaining why. If this was purchased then please include the names of the purchasing sources (e.g., stores, markets, suppliers), if available, as well as any further details about the purchased items (e.g., lot number, source origin, description of appearance) to ensure reproducibility of the analyses. For more information regarding PLOS' policy on materials sharing and reporting, see https://journals.plos.org/plosone/s/materials-and-software-sharing#loc-sharing-materials.

Reply

Comment is well taken. The section was amended as follow.

“An adult mosquito surveillance was conducted in the Ragama Public Health Inspector (PHI) division, located within the District of Gampaha, Sri Lanka in 2018 and captured mosquitoes were transported to the laboratory for mass rearing at the Molecular Medicine Unit, Faculty of Medicine, University of Kelaniya, Sri Lanka. Within the laboratory, only the Ae. aegypti mosquitoes were separated through morphological identification by well-trained entomologists. Eggs laid by a single Ae. aegypti blood-engorged female were used to establish a mosquito colony of Ae. aegypti. Each colony was maintained in 24 x 24 x 24 cm cages with mesh screening on top, under a 12:12 (light:dark) cycle at standard conditions (at 27 ± 2 ∘C and 75 ± 5% humidity)”. 

Comment 4

Please include your tables as part of your main manuscript and remove the individual files. Please note that supplementary tables (should remain/ be uploaded) as separate "supporting information" files.

Reply

Point is well taken. The tables were included as part of our main manuscript. The references list was reviewed. 

Reviewer 1

Comment 1

Authors affiliations:

4"Insect Pest Control Laboratory" instead "Insect Pest Control Subprogramme"

Reply

Sorry but by per co-author instructions, we need to keep Subprogramme since this is the official contact for the IAEA lab.

Comment 2

Abstract

Delete the space before the full-stop at the end of the 1st sentence ‘after Ae. Aegypti’.

Reply

Thank you for your comment. The space was deleted.

Comment 3

Materials and methods

Effects of irradiation on flight ability

Remove the “.” after “9 cm diameter” in the sentence “…..150 pupae of one sex were placed in a 9 cm diameter. Petri dish into which a transparent tube…”

Reply

OK, the sentence was amended as follow: 

“To determine the flight ability of adults, 150 pupae of one sex were placed in a 9 cm diameter Petri dish into which a transparent tube (25 cm in height and 8 cm in diameter) was introduced”.

Comment 4

Statistical analysis

Why “the egg hatch rates were arcsinsqrt-transformed prior to analysis”?

This means that the data was not normally distributed? If the case please precise.

Reply

The data were not normally distributed and so we arcsinsqrt-transformed to meet requirement for analysis. It has been now mentioned at the statistical analysis section. Thank you.

Comment 5

To assess the effects of irradiation on the different parameters (mortality, productivity), the author limits the number of replications to three, however is such studies more replicates are needed to have solid results.

Flies were not available in large number?

Reply

Thank you for your comment. In this experiment 150 pupae/dose/sex were considered and repeated 3 times. We could not do more unfortunately due to the number required even for 3 replicates/dose/sex. 

Comment 6

Mosquitoes -like many insects used for the mass-rearing for SIT- need to be checked for their quality control before using it for SIT. There are several quality control indicators that normally use for this purpose i.e. Mortality/survival, productivity, emergence rate, flight ability, mating competitiveness and insemination rate. In this study, to assess the sterile male mating performance in semi field cage, the author ignore some indicators as the propensity of mating, relative mating index and relative mating performance and limit the study to the competitiveness index.

Or with mosquitoes, is it difficult to evaluate these performance indicators?

Reply 

In this study, we decided to focus on competitiveness, because it is the most important quality control indicator for SIT trials, even if others can be used as well, see Bouyer et al. (2020) (Bouyer, Jérémy, and Marc JB Vreysen. "Yes, irradiated sterile male mosquitoes can be sexually competitive!." Trends in Parasitology (2020). 

.

Comment 7

What means “Hn” and “Hs” in the Friend formula “(Hn-Ho) / (Ho-Hs))*(N/S)”? Please define.

Reply

We welcome the point. The following section was included in the manuscript to address the above comment.

“The formula (Hn-Ho) / (Ho-Hs))*(N/S) was used to calculate the Fried Competitiveness index (C), where Hn and Hs denotes hatch rate from eggs of females mated with untreated and sterile males, respectively, Ho is the observed egg hatch rate when sterile and wild males are competing, N is the number of untreated males and S is the number of sterile males”

Comment 8

The author needs to submit the data analysis details in Rmarkdown file to make the follow up of the data is clear and simple.

Reply

All data files were uploaded. 

Comment 9

Results

1. The figure 2 must be announced in the “Effects of irradiation on flight ability”

Reply

OK, the figure 2 was announced at the ‘Effects of irradiation on flight ability’ section. Thank you. 

Comment 10

In the section “Effects of irradiation on adult survival”, please cheek the figure numbers

Reply

OK, figure numbers were corrected in the section “Effects of irradiation on adult survival”, 

Comment 11

Section “Competitiveness index (C) of sterile male mosquitoes”:

- The tables 1 and 2 are announced but the tables are missing in the document.

Reply

OK, sorry for that, Tables 1 and 2 were included into the document.

Comment 12

The author took about 5:1:1 ratio in the result section but is not mentioned in the methodology section

Reply

OK, the corresponding section was amended as follow.

Laboratory conditions

Four to five days old sterilized and untreated males were introduced into adult mosquito cages (30 cm × 30 cm × 30 cm) together with virgin females of the same age at the 1:1:1, 3:1:1 and 5:1:1 ratios (sterile males: untreated males: virgin females) (n=100). Furthermore, 1:1 ratio of sterile males: virgin females and 1:1 ratio of untreated males: virgin females were introduced into separate cages as sterile and fertile controls. Adult mosquitoes were fed with cattle blood and fed females were counted. Blood-fed females were allowed to lay eggs and the numbers of eggs laid in each egg cup were counted. Collected eggs were hatched and larvae were counted. 

Semi-field conditions

Two experimental ratios were tested in semi-field cages (1.82 m × 1.21 m × 1.21 m) after 4 to 5 days of emergence; sterile males: fertile males: virgin females 1:1:1, 3:1:1 and 5:1 (n=100). Controls were set as 1:1 ratio of sterile males: virgin females and 1:1 ratio of untreated males: virgin females. After a mating period of three days, females were collected into cages (30 cm × 30 cm × 30 cm) and brought to the laboratory. Mosquitoes were blood fed and egg hatch rates were compared as done previously.

Reviewer 2

Comment 1

The manuscript shows a very encouraging beginning of SIT development for Sri Lanka. Great job of the IAEA supporting with technical expertise for this country scientific group. Overall, each of every section in the manuscript accomplishes with academic rules of experimental design; i.e, sample size, repetitions, lab rearing conditions, etc. Field research of mark-release-recapture should be started as soon as possible in order to settle preliminary releases of sterile males in Sri Lanka.

Reply

Authors are thankful for the comment. Field research of mark-release-recapture were completed now and manuscript for mark-release-recapture is being prepared.

---

## [Editor Report · Decision Letter 1]

28 Feb 2022

Development of the Sterile Insect Technique to control the dengue vector Aedes aegypti (Linnaeus) in Sri Lanka

PONE-D-21-28806R1

Dear Dr. Hapugoda,

We’re pleased to inform you that your manuscript has been judged scientifically suitable for publication and will be formally accepted for publication once it meets all outstanding technical requirements.

Kind regards,

Ahmed Ibrahim Hasaballah

Academic Editor

PLOS ONE

---

## [Editor Report · Acceptance letter]

17 Mar 2022

PONE-D-21-28806R1 

Development of the Sterile Insect Technique to control the dengue vector *Aedes aegypti* (Linnaeus) in Sri Lanka 

Dear Dr. Hapugoda:

I'm pleased to inform you that your manuscript has been deemed suitable for publication in PLOS ONE. Congratulations! Your manuscript is now with our production department. 

Kind regards, 

on behalf of

Dr. Ahmed Ibrahim Hasaballah 

Academic Editor

PLOS ONE